# COVID-19’s Impact on Willingness to Be Vaccinated against Influenza and COVID-19 during the 2020/2021 Season: Results from an Online Survey of Canadian Adults 50 Years and Older

**DOI:** 10.3390/vaccines9040346

**Published:** 2021-04-05

**Authors:** Nancy M. Waite, Jennifer A. Pereira, Sherilyn K. D. Houle, Vladimir Gilca, Melissa K. Andrew

**Affiliations:** 1School of Pharmacy, University of Waterloo, Kitchener, ON N2G 1C5, Canada; sherilyn.houle@uwaterloo.ca; 2JRL Research & Consulting Inc., Mississauga, ON L5G 0A3, Canada; jennifer.pereira@jrlresearch.com; 3Institut National de Sante Publique du Quebec, Laval University, Quebec City, QC G1E 7G9, Canada; vladimir.gilca@inspq.qc.ca; 4Department of Medicine (Geriatrics), Dalhousie University, Halifax, NS B3H 2E1, Canada; mandrew@dal.ca

**Keywords:** influenza, COVID, older adults, decision-making, survey

## Abstract

There is considerable overlap in age-related risk factors for influenza and COVID-19. We explored the impact of the pandemic on anticipated influenza and COVID-19 vaccination behaviour in the 2020/2021 season. In May 2020, we conducted online surveys of Canadian adults 50 years and older via a market research panel platform, as part of a series of annual surveys to understand experiences with influenza. Given the current pandemic, respondents were also asked about COVID-19’s impact on their vaccination decision-making for the 2020/2021 season. Of 1001 respondents aged 50–64 years, 470 (47.0%) originally intended on receiving the influenza vaccine and still planned to do so, while 200 (20%) respondents who had planned to abstain now reported willingness to receive the vaccine due to COVID-19. In the 65+ age group, 2525 (72.1%) reported that they had planned to be vaccinated and that COVID-19 had not changed their mind, while 285 individuals (8.1%) reported that they had initially planned to forgo the vaccine but now intended to receive it. Reasons for this change included COVID-19’s demonstration of the devastating potential of viruses; and to protect against influenza, and decrease vulnerability to COVID-19. If the COVID-19 vaccine was available, 69.1% of 50–64 year olds and 79.5% of those 65 years and older reported they would opt to receive it. The COVID-19 pandemic has been a sobering demonstration of the dangers of infectious disease, and the value of vaccines, with implications for influenza and COVID-19 immunization programs.

## 1. Introduction

The SARS-CoV-2 disease 2019 (COVID-19) has caused a global pandemic, with cases exceeding 123 million, and 2,722,000 deaths across the world as of 22 March 2021 [1]. In Canada alone, there have been 943,920 cases, of which 22,695 have resulted in deaths [1]. During influenza season, there is significant concern about how to best protect the public from potential dual circulating viruses, and avoid overwhelming the capacity of our healthcare system. Influenza leads to significant morbidity and mortality annually in Canada, causing over 12,000 hospitalizations and approximately 3500 deaths every year, primarily in older adults [2]. While the national influenza vaccination coverage goal is 80% [3], annual immunization rates of 30–35% and 68–70% are typically reported in those 50–64 years and 65 years and over, respectively [4]. These rates are still insufficient given the transmissibility of this disease (reproduction number for seasonal influenza is often estimated at 1.28) [5] and that these age groups are at increased risk for the complications of influenza [6].

COVID-19 and influenza have common symptoms including fever, cough, rhinitis, headache and myalgia [7,8]. Although the duration and frequency of these symptoms differ between the two diseases, the symptom overlap leads to difficulty in distinguishing between COVID-19 and influenza based on clinical manifestations, requiring laboratory testing for accurate diagnosis [9]. Both viruses are transmitted through direct contact and airborne droplets [10,11]. Multiple common risk factors for severe disease and outcomes have been identified, including increased age (specifically 60 years and over), chronic medical conditions, immunocompromised status and obesity [12,13]. We conducted surveys of older Canadian adults to explore the potential impact of the COVID-19 pandemic on anticipated seasonal influenza vaccination behaviour.

## 2. Materials and Methods

In May 2020, we conducted an online survey of Canadians aged 50–64 years and aged 65 and older. The survey was called *EXamining the knowledge, Attitudes and experiences of Canadian seniors Towards influenza (EXACT)*. This was the third survey in a series to evaluate timely trends in influenza vaccination in Canadians. Previous iterations were administered in April 2017 and August 2019 in order to understand individuals’ experiences during the 2016/2017 [14,15] and 2018/2019 influenza seasons, respectively.

### 2.1. Survey Development

The survey was developed by the study team of influenza researchers and healthcare providers, and comprised multiple choice, and Likert scale of agreement questions. In addition to demographic information, respondents were asked about recent influenza vaccination history, and experiences with influenza and influenza-like illness (sore throat, fever, runny nose and cough). The survey employed adaptive questioning to reduce respondent burden by only presenting certain questions conditionally based on response to previous questions, and were tested for face validity and content validity in a sample of 10 Canadian adults 65 years and over.

Given the current pandemic, we added questions regarding impact of COVID-19 on the respondent’s decision to receive the influenza vaccine for the 2020/2021 season, and the reasons for their decision. We also asked about willingness to receive the COVID-19 vaccine, and preference for vaccine administration setting (for those who responded affirmatively).

Prior to survey implementation, the survey was tested by a study team member (JAP) for technical functionality, including logical adaptive questioning to ensure that all respondents were only shown questions that were relevant to them, based on their previous responses.

### 2.2. Recruitment

Leger Marketing, a Canadian market research firm, disseminated the survey to the two age-based populations through their Online Polling Panel of 400,000 Canadians. This panel was initially recruited through various strategies: over the telephone (60%), referrals/affiliate programs (25%), partner programs (5%), offline recruitment (5%), and social media (5%).

Leger offers financial incentive to panel members based on length of completed surveys. To help ensure data integrity, Leger implements multiple strategies such as data verification questions and analysis of response patterns and times (to identify and eliminate surveys completed unusually quickly, defined as less than one-third the median completion time). Additionally, to prevent biases in question response, multiple choice options were presented in random order.

Leger disseminated the survey directly via an emailed link, to two samples of panel members living in Canada: 50–64 years, and 65 years and older. The link was specific to the email address, and could not be shared with others. The survey had an introduction page that described the study, including the purpose, approximate survey completion time (10–15 min), name and affiliation of the principal investigator (NW), and that Leger would only send the study results to the study team (devoid of all personal identifiers), which would be stored on University of Waterloo servers for 5 years. Completion and submission of the survey implied consent. One to three questions appeared on each survey page, and there were 15–20 pages in total (depending on adaptive questioning). Respondents could change their answers by selecting the “back” button and entering new data; however, once the survey was submitted, the respondent was unable to access the survey. Responses were automatically captured in a database, and only completed surveys could be submitted. Leger sampled proportionately to province population to achieve a sample size of approximately 1000 individuals 50–64 years of age and 3500 individuals 65 years and older. Sample size details have been previously reported [15].

### 2.3. Statistical Analysis

Quantitative data were analyzed overall, and by respondent demographics and influenza vaccination status. Analyses were done using STATA 10.0 (2007, StataCorp, LP, College Station, TX, USA).

### 2.4. Data Analysis

Open-ended responses regarding reasons for opting for or against the influenza vaccine were analyzed thematically. Two reviewers co-coded 20% of the text independently, comparing coding lists throughout the process to maintain consistency. Once they reached consensus on the finalized coding list, they each coded half of the remaining text, conferring as required, when additional codes were needed. As patterns in the data emerged, the codes were grouped into themes.

### 2.5. Ethics Approval

This study was approved by the University of Waterloo Research Ethics Board (ORE#41071 and 40700).

## 3. Results

Between 8 and 29 May 2020, we collected completed surveys from 1001 individuals 50 to 64 years of age (mean age = 57.1; 51.1% female; 48.5% with chronic conditions; median completion time = 9 min) and 3500 individuals 65 years and older (mean age = 71.4; 54.7% female; 64.2% with chronic conditions; median completion time = 13 min) with representation from all ten Canadian provinces (Table 1). Overall, the survey response rate was 31% for 50–64 year olds, and 42% for those 65 years and older.

### 3.1. COVID-19’s Impact on Willingness to Receive the Influenza Vaccine

Of those aged 50–64 years, 780 (77.9%) reported their influenza vaccination decision-making for the 2020/2021 season was unaffected by COVID-19: 470 (47.0%) still intended on being vaccinated, and 310 (31.0%) still did not (Figure 1). However, 200 (20.0%) respondents reported that COVID-19 made them more likely to be vaccinated against influenza, despite previous reservations. Of this subset, 92.0% had not been vaccinated against influenza in 2019/2020.

When respondents aged 50 to 64 years who had not originally intended to receive the influenza vaccine but then reported that they changed their mind due to COVID-19, were compared with those who did not alter their decision to abstain, we did not find any significant differences in demographics include age, sex, location of residence and chronic conditions (Table 2).

In the 65+ age group, 2525 (72.1%) still planned on being vaccinated and 662 (18.9%) still planned to abstain (Figure 1). Two-hundred eighty-five individuals (8.1%) reported that COVID-19 made them more likely to opt for influenza vaccine (92.3% of whom were unvaccinated in 2019/2020).

We compared demographics for respondents 65 years and older who had originally intended to abstain from the influenza vaccine but reported that COVID-19 made them more likely to receive it, with those who remained unwilling to receive the vaccine (Table 2). Those who changed their mind were more likely to be female (62.1% vs. 56.6%) although this difference was not statistically significant. They were significantly more likely to have hypotension or hypertension (48.1% vs. 33.8%; *p* < 0.001) and less likely to have no chronic conditions (37.9% vs. 48.0%; *p* < 0.01).

Across both age groups, respondents reported several reasons for this change, from which we identified the following overarching themes:COVID-19 had demonstrated the devastating potential of viruses, which caused fear among those who were previously unconcerned and sparked the realization that they may not be invincible to the severe outcomes of influenza, due to their age and other risk factors;Sentiment that this would be a risky season due to both influenza and COVID-19 circulating, and that by preventing influenza they could reduce confusion about what illness they have (thereby, perhaps facilitating treatment), if they indeed acquired an infection;To avail of all viral protection, regardless of personal beliefs on the effectiveness and necessity of influenza vaccines;To protect against influenza, and thereby be healthier during the winter season and potentially less vulnerable to COVID-19 and its clinical consequences; andHopefulness that given the common symptoms and transmission of influenza and COVID-19, the influenza vaccine could offer some protection against COVID-19, given that a COVID-19 vaccine was unlikely to be available by the winter season.

A minority (2.1% of 50–64 year olds and 0.9% of 65+) reported being less likely to receive the influenza vaccine due to COVID-19. The two primary reasons were fear of being exposed to COVID-19 in an influenza vaccination setting, and because they planned on still social distancing during the influenza season so anticipated little risk of exposure to the virus.

### 3.2. Willingness to Receive a Vaccine for COVID-19

We asked respondents whether they would be willing to receive a vaccine against COVID-19 if one was available. The majority of respondents in both populations (more than for influenza vaccine) reported that they would opt for the vaccine (69.1% of those 50–64 years, and 79.5% of those 65 years and older) (Table 3).

Respondents 50–64 years old who reported that they would opt for the COVID-19 vaccine were significantly more likely to be male than those who were unsure about the vaccine, and significantly more likely to have at least one chronic condition (*p* < 0.05; Table 4). Of the 580 50–64 year-old respondents who were not vaccinated against influenza in the 2019/2020 season, 340 (58.6%) would opt for the COVID-19 vaccine. Similarly, 611 of the 1074 individuals 65 years and older who were not vaccinated against influenza in 2019/2020 (56.9%) reported that they would opt for the COVID-19 vaccine. Respondents 65 years old and older who reported that they would opt for the COVID-19 vaccine were significantly more likely to be male than those who reported they would abstain, and significantly more likely to have at least one chronic condition, heart disease and high or low blood pressure (*p* < 0.05).

When those who indicated they would receive a COVID-19 vaccine were asked at which setting would they prefer to receive it, the majority of respondents in both groups chose their family physician office, followed by pharmacy, workplace (for those 50–64 years), and public health clinics (Figure 2).

## 4. Discussion

To understand whether COVID-19 has impacted willingness to receive the influenza vaccine, we conducted a survey of Canadian adults 50 years and older, given that this population is more likely to have risk factors for complications. In the 50–64 year group, 67% reported that they were going to receive the influenza vaccine in the 2020/2021 season; this comprised 47% who had previously intended on receiving the influenza vaccine, and for whom COVID-19 had not impacted their decision, and an additional 20% of respondents who had previously decided to forgo the influenza vaccine but were now opting for it due to the impact of COVID-19. In those 65 years and older, 72.1% had previously planned on being vaccinated and still intended to do so while 8.1% reported that they had originally intended to abstain from the influenza vaccine but that COVID-19 made them more likely to opt for it. Across both age groups, the majority of those who reported that the pandemic had impacted their influenza vaccine decision-making and that they were now willing to be vaccinated, had not been vaccinated in the previous season. Respondents reported several reasons for this change including having witnessed the devastating potential of viruses, to protect themselves against viruses through all available means, and on the chance that the influenza vaccine will offer some level of protection against COVID-19. We also noted that an even greater majority of respondents—69.1% of 50–64 year olds and 79.5% of those 65 years and older—reported they would receive a COVID-19 vaccine, if available. These results indicate that to many, including those previously vaccine-hesitant, the COVID-19 pandemic has been a sobering demonstration of the dangers of infectious disease, and the value of vaccines [16]. The fact that more older adults reported that they would want a COVID-19 vaccine than an influenza one suggests that they may perceive the risks of COVID-19 to be greater than influenza, though this was not specifically asked in our survey.

We also noted a small subset of respondents in both age groups who were previously intending on receiving the influenza vaccine in the 2020/2021 season but who had decided against it, due to COVID-19. The predominant reasons given for this change in decision were directly tied to their perceived lower risk of influenza in itself (due to lower exposure to others because of social distancing) and in relation to their perceived risk of acquiring COVID-19 while being vaccinated against influenza. While it is reassuring to note that only a small percentage of respondents were dissuaded from receiving the influenza vaccine, our study demonstrates that for some individuals, receiving the influenza vaccine is not necessarily a simple annual routine or habit but a careful weighing of risks and benefits.


Our findings are supported by another recent national online survey of 1912 Canadians 18 years and older which found that 57% of respondents reported that they would receive the influenza vaccine this season, a notable increase from the 45% who indicated they received the vaccine in the past season [17]. Of these respondents, 34% indicated that they were more likely to get the influenza vaccine this year, due to COVID-19. Similar to that survey, our baseline data on influenza and COVID-19 vaccine intent can be compared to actual vaccination behaviour in the 2020/2021 influenza season and beyond to further understand vaccine decision-making factors. In addition, the new knowledge that our study provides on why a pandemic so greatly increases interest in getting the vaccine may provide guidance in how to address vaccine hesitancy in non-pandemic times. There are unique circumstances involved with assessing perceptions of influenza during a pandemic, given the constantly evolving nature of the emergent infection, the new vaccine-related research and data that is being reported on daily, and the changing fears of the public. Preventing disease is top of mind for society to an extent that is not likely to be representative of non-pandemic times. It therefore becomes important to constantly take the pulse of the public, and understand vaccination-related perceptions and concerns in order to be able to address them. For example, a subset of respondents indicated that their newfound knowledge on the consequences of virus infections increased their willingness to receive the influenza vaccine. Therefore, public health messaging could be more focused on the impact of diseases, with respect to how they affect health, associated long-term and severe complications, and potential effects on the ability to conduct activities of daily living. Additionally, we found that a small subset of respondents were concerned about contracting COVID-19 at influenza vaccination clinics; therefore, in addition to usual strategies for addressing vaccine hesitancy, healthcare professionals likely need to reassure the public about the various physical distancing and sanitation measures that are undertaken to reduce the risk of transmission. While we asked respondents about the reasons for their change in decision, we did not specifically ask about the source of information that led to the change. Knowing whether it was primarily related to media coverage, discussions with family or friends, or scientific literature would be an important next step in informing new messaging about vaccine preventable infections.

Respondents were asked about their preference for COVID-19 vaccination setting, and the majority in both age groups reported a preference for receiving the vaccine in their physician’s office. This is likely attributed to a number of factors including the vaccine being new to the public, increasing inclination to receive it from a medical professional whom they are already familiar with and trust. Respondents in the 50–64 year group were more likely to opt to receive the COVID vaccine in a pharmacy than the older age group, which may be due to the convenience associated with a pharmacy setting being more of a priority for those in the workforce. However, in several Canadian provinces, pharmacists have only recently had their scope of practice expanded to include immunization [18], so many respondents may not yet be accustomed to receiving vaccinations in a pharmacy setting.

This study has several strengths including the large number of respondents, inclusion of all provinces, and samples similar to the Canadian 50–64 year and senior populations in terms of sex distribution and chronic illness prevalence, and influenza vaccination rates in 2019/2020 [19,20,21].

Our study had also some limitations. As is true of all surveys, it is possible that those who responded may have different opinions from those who did not. Given our online sampling frame, we likely excluded Canadians who are very ill or cognitively impaired, and may have oversampled from those with strong views—negative or positive—on vaccination or COVID-19. Furthermore, our survey was conducted before details about the current approved and candidate COVID-19 vaccines were available, and so responses reflect a general sentiment about COVID-19 vaccine willingness at that point in time and are not specific to subsequent knowledge of actual available vaccines.


## 5. Conclusions

In summary, our study has generated solid baseline data regarding both influenza and COVID-19 vaccine uptake intent. Our findings indicate that the current pandemic has provided strong reasons for some older adults to be accepting of the influenza vaccine; such knowledge may prove useful in directing public health efforts to combat vaccine hesitancy. Future studies are needed to determine whether respondents’ reported opinions translated to actual behaviour—given the implications to influenza immunization programs, it will be important to understand whether any such changes are temporary or will be maintained during influenza seasons in non-pandemic times.


## Figures and Tables

**Figure 1 vaccines-09-00346-f001:**
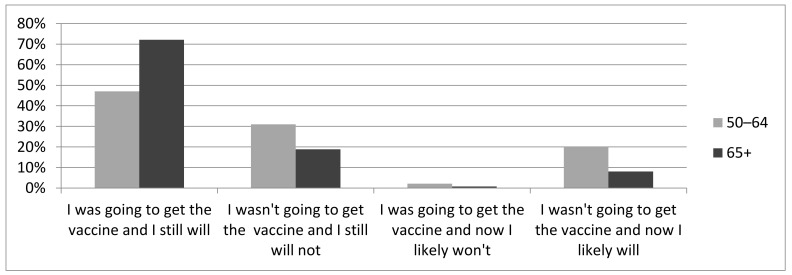
Impact of COVID-19 on decision to receive or abstain from influenza vaccine.

**Figure 2 vaccines-09-00346-f002:**
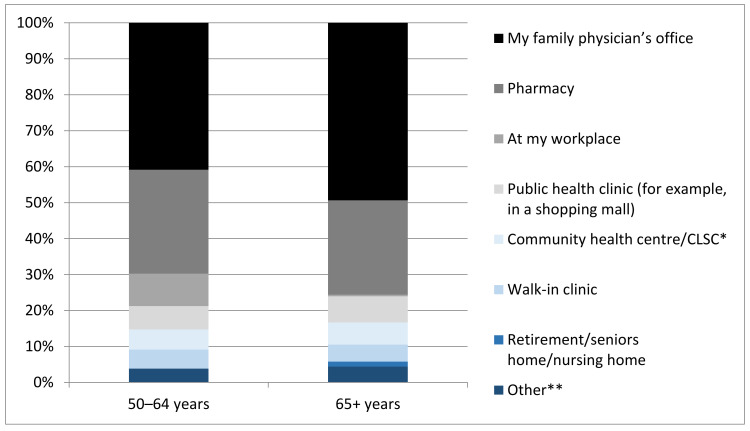
Preferred location to receive COVID-19 vaccine. * CLSC = Centre local de services communautaires (form of community health centre in Quebec). ** Other = undecided, no preference, home.

**Table 1 vaccines-09-00346-t001:** Baseline characteristics of respondents.

	50–64 Years	65 Years and Older
	*n* = 1001	*n* = 3500
Characteristic	*n* (%)	*n* (%)
**Age**		
Mean age (yrs), SD	57.1, 4.1	71.4, 5.3
50–54 years	325 (32.5)	
55–59 years	332 (33.2)	
60–64 years	344 (34.4)	
65–74 years		2663 (76.1)
75+		837 (23.9)
**Biological Sex**		
Male	489 (48.9)	1586 (45.3)
Female	512 (51.1)	1913 (54.7)
Other	0 (0)	1 (0.03)
**Province**		
British Columbia	135 (13.5)	503 (14.4)
Alberta	105 (10.5)	296 (8.5)
Saskatchewan	26 (2.6)	104 (3.0)
Manitoba	37 (3.7)	120 (3.4)
Ontario	385 (38.5)	1321 (37.7)
Quebec	240 (24.0)	883 (25.2)
New Brunswick	24 (2.4)	85 (2.4)
Nova Scotia	28 (2.8)	115 (3.3)
Prince Edward Island	5 (0.5)	14 (0.4)
Newfoundland and Labrador	16 (1.6)	59 (1.7)
**Location of residence**		
City	608 (60.8)	2009 (57.4)
Town	324 (32.4)	1174 (33.5)
Village	69 (6.9)	317 (9.1)
**Chronic condition(s)**		
None	516 (51.5)	1252 (35.8)
Diabetes	144 (14.4)	600 (17.1)
Heart disease	47 (4.7)	345 (9.9)
Asthma or chronic lung disease other than COPD	104 (10.4)	347 (9.9)
Blood disorders (not including high or low blood pressure)	22 (2.2)	97 (2.8)
High or low blood pressure	301 (30.1)	1575 (45.0)
COPD	21 (2.1)	198 (5.7)
Cancer	14 (1.4)	175 (5.0)
Neurological disorders	32 (3.2)	86 (2.5)
Kidney disease	11 (1.1)	94 (2.7)
Significant trouble with memory	13 (1.3)	37 (1.1)
Liver disease	10 (1.0)	34 (1.0)
Have had a transplant AND/OR have an immunosuppressive condition AND/OR taking an immunosuppressive medication	19 (1.9)	44 (1.3)
HIV/AIDS	8 (0.8)	7 (0.2)

**Table 2 vaccines-09-00346-t002:** Characteristics of respondents who originally did not intend to receive the influenza vaccine, by age group.

	50–64 Years		65 Years and Older	
	Changed Decision *n* = 200	Did not Change Decision *n* = 310	Changed Decision *n* = 285	Did not Change Decision *n* = 662
Characteristic	*n* (%)	*n* (%)	*n* (%)	*n* (%)
**Age**				
Mean age (yrs)	57.0	56.7	70.1	70.1
**Biological Sex**				
Male	98 (49.0)	145 (46.8)	108 (37.9)	287 (43.4)
Female	102 (51.0)	165 (53.2)	177 (62.1)	375 (56.6)
Other				
**Location of residence**				
City	125 (62.5)	176 (56.8)	146 (51.2)	336 (50.8)
Town	59 (29.5)	113 (36.5)	108 (37.9)	259 (39.1)
Village	16 (8.0)	21 (6.8)	31 (10.9)	67 (10.1)
**Chronic condition (s)**				
None	118 (59.0)	187 (60.3)	108 (37.9) *	318 (48.0) *
Diabetes	20 (10.0)	35 (11.3)	36 (12.6)	84 (12.7)
Heart disease	9 (4.5)	12 (3.9)	29 (10.2)	35 (5.3)
Asthma or chronic lung disease other than COPD	15 (7.5)	18 (5.8)	20 (7.0)	52 (7.9)
Blood disorders (not including high or low blood pressure)	4 (2.0)	6 (1.9)	5 (1.8)	11 (1.7)
High or low blood pressure	53 (26.5)	80 (25.8)	137 (48.1) *	224 (33.8) *
COPD	3 (1.5)	3 (1.0)	16 (5.6)	18 (2.7)
Cancer	4 (2.0)	4 (1.3)	12 (4.2)	28 (4.2)
Neurological disorders	7 (3.5)	10 (3.2)	6 (2.1)	14 (2.1)
Kidney disease	1 (0.5)	1 (0.3)	7 (2.5)	11 (1.7)
Significant trouble with memory	5 (2.5)	3 (0.9)	2 (0.7)	6 (0.9)
Liver disease	1 (0.5)	2 (0.6)	6 (2.1)	2 (0.3)
Have had a transplant AND/OR have an immunosuppressive condition AND/OR taking an immunosuppressive medication	3 (1.5)	2 (0.6)	2 (0.7)	0 (0)
HIV/AIDS	0 (0)	0 (0)	0 (0)	0 (0)

* *p* < 0.05.

**Table 3 vaccines-09-00346-t003:** If there was a COVID-19 vaccine, would you receive it?

	50–64 Years *n* = 1001 *n* (%)	65+ Years *n* = 3500 *n* (%)
Yes	692 (69.1)	2782 (79.5)
No	113 (11.3)	196 (5.6)
I don’t know	196 (19.6)	522 (14.9)

**Table 4 vaccines-09-00346-t004:** Characteristics of respondents based on reported willingness to reeive the COVID-19, by age group.

	50–64 Years			65 Years and Older		
Decision to Receive COVID-19 Vaccine	Yes (*n* = 692)	No (*n* = 113)	I don’t Know (*n* = 196)	Yes (*n* = 2782)	No (*n* = 196)	I don’t Know (*n* = 522)
Characteristic	*n* (%)	*n* (%)	*n* (%)	*n* (%)	*n* (%)	*n* (%)
**Age**						
Mean age (yrs)	57.4	56.3	56.5	71.5	70.6	70.8
**Biological Sex**						
Male	361 (52.2) *	49 (43.4)	79 (40.3) *	1322* (47.5)	70 (35.7) *	194 (37.2)
Female	331 (47.8) *	64 (56.6)	117 (59.7) *	1459 (52.4) *	126 (64.3) *	328 (62.8)
Other	0 (0)	0 (0)	0 (0)	1 (0.03)	0 (0)	0 (0)
**Chronic condition(s)**						
None	343 (49.6) *	69 (61.1) *	104 (53.1)	951 (34.2) *	99 (50.5) *	202 (38.7)
Diabetes	97 (14.0)	29 (17.7)	27 (13.8)	492 (17.7)	27 (13.8)	81 (15.6)
Heart disease	32 (4.6)	7 (6.2)	8 (4.1)	299 (10.7) *	7 (3.6) *	39 (7.5)
Asthma or chronic lung disease other than COPD	75 (10.8)	12 (10.6)	17 (8.7)	271 (9.7)	20 (10.2)	56 (10.7)
Blood disorders (not including high or low blood pressure)	13 (1.9)	4 (3.5)	5 (2.6)	77 (2.8)	4 (2.0)	16 (3.1)
High or low blood pressure	218 (31.5)	32 (28.3)	51 (26.0)	1295 (46.5) *	65 (33.2) *	215 (41.2)
COPD	16 (2.3)	2 (1.8)	3 (1.5)	167 (6.0)	5 (2.6)	26 (5.0)
Cancer	11 (1.6)	0 (0)	3 (1.6)	144 (5.2)	8 (4.1)	23 (4.4)
Neurological disorders	22 (3.2)	4 (3.5)	6 (3.1)	63 (2.3)	7 (3.6)	16 (3.1)
Kidney disease	7 (1.0)	0 (0)	4 (2.0)	73 (2.6)	1 (0.5)	20 (3.8)
Significant trouble with memory	11 (1.6)	0 (0)	2 (1.0)	26 (0.9)	3 (1.5)	8 (1.5)
Liver disease	7 (1.0)	3 (2.7)	0 (0)	30 (1.1)	2 (1.0)	2 (0.4)
Have had a transplant AND/OR have an immunosuppressive condition AND/OR taking an immunosuppressive medication	14 (2.0)	1 (1.0)	4 (2.0)	37 (1.3)	1 (0.5)	6 (1.1)
HIV/AIDS	8 (1.1)	0 (0)	0 (0)	7 (0.3)	0 (0.)	0 (0)

* *p* < 0.05.

## Data Availability

The datasets generated and analyzed during the current study are not publicly available given that study participants did not consent for their individual data to be shared (aggregate data sharing only), and the possibility that a subset of participants may be identified from their responses which would compromise anonymity.

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
