# Peer review of "COVID-19’s Impact on Willingness to Be Vaccinated against Influenza and COVID-19 during the 2020/2021 Season: Results from an Online Survey of Canadian Adults 50 Years and Older"

_vaccines, 2021, doi:10.3390/vaccines9040346_

Round 1

Reviewer 1 Report

Thank you for the opportunity to review this manuscript. In this manuscript, Waite et al report that the impact of the COVID-19 pandemic on willingness to be vaccinated against influenza virus and SARS-CoV2 in Canada. This manuscript indicated that the COVID-19 pandemic impacts the acceptance rates to the influenza vaccines in older people. This manuscript is well-written and structured. However, there are a few concerns that need to be clarified before publication stated below.

  1. Page 2, lines 41-44: Please address the relationship between vaccine coverage rates (or herd immunity threshold) and R0 (basic reproduction number).
  2. Page 2, lines 50-51: Did you perform any cross-tabulation analysis?
  3. Figure 1: Is there any data or suggestion about the reason why some people changed their minds? e.g., based on what information or which source (SNS? family? friends? old media?). Please at least discuss this.
  4. Please carefully go through the manuscript to check the word "COVID-19" is appropriate; maybe "SARS-CoV2" is accurate somewhere.

Author Response

Dear Editors,

Thank you for your reviewers’ thorough and thoughtful re-review of our manuscript entitled, “COVID-19’s impact on willingness to be vaccinated against influenza and COVID-19 during the 2020/2021 season: results from an online survey of Canadian adults 50 years and older” (Vaccines-1149926) which was submitted for consideration by Vaccines. We appreciate your suggestions and believe that we have satisfactorily responded to your questions below, including adding more information where helpful and making slight modifications to keep within the word limit. We look forward to hearing from you regarding your decision on our manuscript.

Sincerely,

Nancy Waite

Associate Director

School of Pharmacy, University of Waterloo

10A Victoria St. S. 

Kitchener, Ontario 

N2G 1C5

On behalf of Jennifer Pereira, Sherilyn Houle, Vladimir Gilca and Melissa Andrew

Reviewer 1 Comments

Response

  1. Page 2, lines 41-44: Please address the relationship between vaccine coverage rates (or herd immunity threshold) and R0 (basic reproduction number).

We have revised this sentence to state: “These rates are still insufficient given the transmissibility of this disease (reproduction number for seasonal influenza is often estimated at 1.28) [5] and that these age groups are at increased risk for the complications of influenza.[6]”

2.       Page 2, lines 50-51: Did you perform any cross-tabulation analysis?

Given that our COVID-19-related questions were only asked during the recent 2019/2020 survey we did not perform cross-tabulation analysis based on year for this particular paper.

3.      Figure 1: Is there any data or suggestion about the reason why some people changed their minds? e.g., based on what information or which source (SNS? family? friends? old media?). Please at least discuss this.

This is an excellent point. While we did ask about the reason for the change in mind, we did not ask specifically about the source of that information so are unable to comment on whether it was the result of personal experience, family/friends, healthcare providers, media, etc. We have added the following sentence to the end of the 3d paragraph of the Discussion:

“While we asked respondents about the reasons for their change in decision, we did not specifically ask about the source of information that led to the change. Knowing whether it was primarily related to media coverage, discussions with family or friends, or scientific literature would be an important next step in informing new messaging about vaccine preventable infections.”

4.      Please carefully go through the manuscript to check the word "COVID-19" is appropriate; maybe "SARS-CoV2" is accurate somewhere.

We have made the suggested change throughout the manuscript, where appropriate.

Reviewer 2 Report

This study and manuscript focus on timely and highly relevant topics related to influenza and COVID-19 vaccination, including whether COVID-19 has impacted flu vaccination intentions. The survey was well designed and implemented, with a sound description of the survey method provided. The findings provide useful insights into 1) reasons why respondents who otherwise were not planning to get an influenza vaccination decided to do so and 2) the number/percentage of people who decided not to get an influenza vaccination as a result of the COVID-19 pandemic. With respect to the latter, the survey's results illustrate that only a very small number of people reported avoiding getting a flu vaccination because of COVID-19 concerns. Overall, this manuscript was well organized and written. My suggestions for further strengthening the manuscript are:

1) It would be helpful to provide, if possible, more information regarding the 50-64 year olds and 65+ respondents who had changed their position on getting a flu vaccination. For example, were there any characteristics associated with those respondents, particularly with respect to biological sex, location of residence, and major chronic health conditions?

2) It would also be helpful if more information could be provided regarding the 50-64 years olds and 65+ respondents who indicated they were not going to get an influenza vaccination and still do not plan to. It would similarly help to do so with respect to those planning and not planning to get a COVID-19 vaccination. 

3) In the first sentence of the Introduction, it would be more accurate to state "The SARS-CoV-2 virus has caused a global pandemic. . ." (i.e., it is the virus, not the disease/illness, that has caused a pandemic).

4) It would be helpful in the Discussion to highlight how few people were dissuaded from getting a flu vaccination as a result of COVID-19. This is mentioned in the Discussion, but given there was speculation that large numbers of people might forego flu vaccination because of perceived risk of acquiring COVID-19, the findings from this study are very reassuring.

5) The statement in lines 204-207 is poorly worded and quite vague. With respect to wording, the sentence is too long and it is unclear what "diseases" are being referred to. Also, the advice to make public health messaging "more explicit" is too general. More explicit how? And why? 

6) Line 230, the sentence would be more accurate if "offered" were replaced with "provided." 

7) Lines 229-231 - while it is true that "such knowledge may prove useful in directing public health efforts to combat vaccine hesitancy," the Discussion section currently does not explicitly or well describe how the findings could be used by public health programs to combat COVID-19 vaccine hesitancy.

Author Response

Dear Editors,

Thank you for your reviewers’ thorough and thoughtful re-review of our manuscript entitled, “COVID-19’s impact on willingness to be vaccinated against influenza and COVID-19 during the 2020/2021 season: results from an online survey of Canadian adults 50 years and older” (Vaccines-1149926) which was submitted for consideration by Vaccines. We appreciate your suggestions and believe that we have satisfactorily responded to your questions below, including adding more information where helpful and making slight modifications to keep within the word limit. We look forward to hearing from you regarding your decision on our manuscript.

Sincerely,

Nancy Waite

Associate Director

School of Pharmacy, University of Waterloo

10A Victoria St. S. 

Kitchener, Ontario 

N2G 1C5

On behalf of Jennifer Pereira, Sherilyn Houle, Vladimir Gilca and Melissa Andrew

Reviewer 2 Comments

Response

1.      It would be helpful to provide, if possible, more information regarding the 50-64 year olds and 65+ respondents who had changed their position on getting a flu vaccination. For example, were there any characteristics associated with those respondents, particularly with respect to biological sex, location of residence, and major chronic health conditions?

We have added the following to Section 3.1:

“When respondents aged 50-64 years who had not originally intended to receive the influenza vaccine but then reported that they changed their mind due to COVID-19, were compared with those who did not alter their decision to abstain, we did not find any significant differences in demographics including age, sex, location of residence or chronic conditions.”

“We compared demographics for respondents 65 years and older who had originally intended to abstain from the influenza vaccine but reported that COVID-19 made them more likely to receive it, with those who remained unwilling to receive the vaccine. Those who changed their mind were more likely to be female (62.1% vs. 56.6%) although this difference was not statistically significant. They were significantly more likely to have hypertension (48.1% vs. 33.8%; p <0.001) and less likely to have chronic conditions (37.9% vs. 48.0%; p<0.01).” 

2.      It would also be helpful if more information could be provided regarding the 50-64 years olds and 65+ respondents who indicated they were not going to get an influenza vaccination and still do not plan to. It would similarly help to do so with respect to those planning and not planning to get a COVID-19 vaccination.

We have added Table 2 and Table 4 to describe the characteristics of these sub-populations.

3.      In the first sentence of the Introduction, it would be more accurate to state "The SARS-CoV-2 virus has caused a global pandemic. . ." (i.e., it is the virus, not the disease/illness, that has caused a pandemic).

We agree and have made the suggested change to the first sentence of the Introduction.

4.      It would be helpful in the Discussion to highlight how few people were dissuaded from getting a flu vaccination as a result of COVID-19. This is mentioned in the Discussion, but given there was speculation that large numbers of people might forego flu vaccination because of perceived risk of acquiring COVID-19, the findings from this study are very reassuring.

Thank you for this point. We have added the following sentence to the 2nd paragraph of the Discussion:

“While it is reassuring to note that only a small percentage of respondents were dissuaded from receiving the influenza vaccine, our study demonstrates that for some individuals, receiving the influenza vaccine is not necessarily a simple annual routine or habit but a careful weighing of risks and benefits.” 

5.      The statement in lines 204-207 is poorly worded and quite vague. With respect to wording, the sentence is too long and it is unclear what "diseases" are being referred to. Also, the advice to make public health messaging "more explicit" is too general. More explicit how? And why? 

We have revised this sentence to state:

“For example, a subset of respondents indicated that their newfound knowledge on the consequences of viral infections increased their willingness to receive the influenza vaccine. Therefore, public health messaging could be more focused on the impact of diseases, with respect to how they affect health, associated long-term and severe complications, and potential effects on the ability to conduct activities of daily living.”

6.      Line 230, the sentence would be more accurate if "offered" were replaced with "provided." 

We have made the suggested change to the second sentence of the Conclusion.

7.      Lines 229-231 - while it is true that "such knowledge may prove useful in directing public health efforts to combat vaccine hesitancy," the Discussion section currently does not explicitly or well describe how the findings could be used by public health programs to combat COVID-19 vaccine hesitancy.

We have added the following paragraph into the 3rd paragraph of the Discussion:

“There are unique circumstances involved with assessing perceptions of influenza during a pandemic, given the constantly evolving nature of the emergent infection, the new vaccine-related research and data that is being reported on daily, and the changing fears of the public. Preventing disease is top of mind for society to an extent that is not likely to be representative of non-pandemic times. It therefore becomes important to constantly take the pulse of the public, and understand vaccination-related concerns in order to be able to address them. For example, we found that a small subset of respondents were concerned about contracting COVID-19 at influenza vaccination clinics; therefore, in addition to usual strategies for addressing vaccine hesitancy, healthcare professionals likely need to reassure the public about the various physical distancing and sanitation measures that are undertaken to reduce the risk of transmission.”

Reviewer 3 Report

This survey on vaccine acceptance addressing both flu and covid immunization is timely and informative.  It confirms observations by others, that flu vaccine acceptance was up this year, seemingly  related to covid concerns.  The detail in some comments (e.g. why some small portion of respondents would NOT get flu vaccine this year, despite usually getting it) is helpful.  The report is clearly presented, the information can be helpful to vaccine promotion in the future. 

Specific comments: 

  1. Introduction: it would be reasonable to note the month/year in the first paragraph, as this article may be referred to in the future and readers will need to know the Covid statistics pertained to a specific time. (E.g. 97 million cases as of when...).  
  2. Methods: there are some terms specific to survey research which could be explained, e.g. 'adaptive questioning" and "logical adaptive questioning".  All readers may not be survey research experts and some might appreciate explanations of these terms, and/or examples from the survey. 
  3. Results: It would be helpful to have some comparison with the Canadian population; to what extent do the 400,000 persons in the Online Polling Panel represent Canada? To what extent do the 50+ year olds who were included, and who responded, represent that age group?  This information would be important for future vaccine promotion. 
  4. Discussion: on line 197 there is reference to another survey, this is referenced but not explained. It would be helpful to briefly put that survey in context (e.g. in another on line survey conducted as part of X study...).  

Congratulations on an interesting and well presented survey.

Author Response

Dear Editors,

Thank you for your reviewers’ thorough and thoughtful re-review of our manuscript entitled, “COVID-19’s impact on willingness to be vaccinated against influenza and COVID-19 during the 2020/2021 season: results from an online survey of Canadian adults 50 years and older” (Vaccines-1149926) which was submitted for consideration by Vaccines. We appreciate your suggestions and believe that we have satisfactorily responded to your questions below, including adding more information where helpful and making slight modifications to keep within the word limit. We look forward to hearing from you regarding your decision on our manuscript.

Sincerely,

Nancy Waite

Associate Director

School of Pharmacy, University of Waterloo

10A Victoria St. S. 

Kitchener, Ontario 

N2G 1C5

On behalf of Jennifer Pereira, Sherilyn Houle, Vladimir Gilca and Melissa Andrew

Reviewer 3 Comments

Response

  1. Introduction: it would be reasonable to note the month/year in the first paragraph, as this article may be referred to in the future and readers will need to know the Covid statistics pertained to a specific time. (E.g. 97 million cases as of when...).  

We agree, and have updated the numbers and added in “as of March 22, 2021” to the sentence.

  1.  Methods: there are some terms specific to survey research which could be explained, e.g. 'adaptive questioning" and "logical adaptive questioning".  All readers may not be survey research experts and some might appreciate explanations of these terms, and/or examples from the survey. 

We have modified the 3rd sentence in Methods, Survey Development: “The survey employed adaptive questioning to reduce respondent burden by only presenting certain questions conditionally based on response to previous questions.”

We have modified the last sentence in Methods, Survey Development to: “Prior to survey implementation, the survey was tested by a study team member (JAP) for technical functionality, including logical adaptive questioning to ensure that all respondents were only shown questions that were relevant to them, based on their previous responses.”

  1. Results: It would be helpful to have some comparison with the Canadian population; to what extent do the 400,000 persons in the Online Polling Panel represent Canada? To what extent do the 50+ year olds who were included, and who responded, represent that age group?  This information would be important for future vaccine promotion. 

We have included the following section in the Discussion: “This study has several strengths including the large number of respondents, inclusion of all provinces, and samples similar to the Canadian 50-64 year and senior populations in terms of sex distribution and chronic illness prevalence, and influenza vaccination rates in 2019/2020.[18-20]”

1.      Discussion: on line 197 there is reference to another survey, this is referenced but not explained. It would be helpful to briefly put that survey in context (e.g. in another on line survey conducted as part of X study...).  .

We have added the following sentence to the 3rd last paragraph of the Discussion:

“Our findings are supported by another recent national online survey of 1,912 Canadians 18 years and older which found that 57% of respondents reported that they would receive the influenza vaccine this season, a notable increase from the 45% who indicated they received the vaccine in the past season.[16]”
